# Controversial Issues Related to Dye Adsorption on Clay Minerals: A Critical Review

**DOI:** 10.3390/molecules28196951

**Published:** 2023-10-06

**Authors:** Juraj Bujdák

**Affiliations:** 1Department of Physical and Theoretical Chemistry, Faculty of Natural Sciences, Comenius University in Bratislava, 842 15 Bratislava, Slovakia; juraj.bujdak@uniba.sk; Tel.: +421-2-9014-9602; 2Institute of Inorganic Chemistry, Slovak Academy of Sciences, 845 36 Bratislava, Slovakia

**Keywords:** environmental, adsorption mechanism, adsorption isotherm, clays, waste adsorption

## Abstract

This critical review points out the most serious and problematic issues to be found in the literature on the adsorption of dyes on clay minerals. The introduction draws attention to the fundamental problems, namely the insufficient characterization of adsorbents, the influence of impurities on the adsorption of dyes, and the choice of inappropriate models for the description of the very complex systems that clay minerals and their systems represent. This paper discusses the main processes accompanying adsorption in colloidal systems of clay minerals. The relationship between the stability of the colloidal systems and the adsorption of dye molecules is analyzed. The usual methodological procedures for determining and evaluating the adsorption of dyes are critically reviewed. A brief overview and examples of modified clay minerals and complex systems for the adsorption of organic dyes are summarized. This review is a guide for avoiding some faults in characterizing the adsorption of organic dyes on clay minerals, to improve the procedure for determining adsorption, to evaluate results correctly, and to find an appropriate theoretical interpretation. The main message of this article is a critical analysis of the current state of the research in this field, but at the same time, it is a guide on how to avoid the most common problems and mistakes.

## 1. Introduction

To date, thousands of papers have been published dealing with the adsorption of organic dyes onto clays, clay minerals, modified clays or clay minerals, derived materials, and related multicomponent systems and composites of various types. Since the first reports appeared sometime in the middle of the 20th century, e.g., [1], a huge number of papers have been published, and new ones are still being published today. Very few studies dealing with dye adsorption are based on a thorough analysis of older published papers. Many papers have little added value compared to older contributions to the literature and often represent only duplicates or very similar studies to previously published reports. In most cases, the only difference is the use of different clay or clay mineral samples. Comparisons with older, similar studies and consistent analyses are rather rare.

This review paper has the ambition to point out the most serious contradictions of the studies published in this field, starting with basic physical theories, pointing out problems with the applied methodology of experiments, data acquisition, and processing, and last but not least, with the interpretation of results and the contributions to related scientific disciplines. The complexity of the problems is highlighted and the overall analysis opens new avenues for future research.

In addition to the original research papers published in this field, numerous review articles have been published analyzing the adsorption of dye molecules on clay minerals and attempting to systematically organize the knowledge on this subject [2,3,4,5,6,7,8,9,10,11,12,13,14,15,16,17,18,19,20,21]. These attempts have been made to find a system of laws between the nature and structure of the interacting species and the characteristics of the adsorption process. The analyses were mostly based on knowledge that was obtained from collecting data from numerous publications, trying to explain adsorption by applying physical laws and interpreting them based on the structure and properties of the adsorbents and the chemical entities involved. The objectives of some review articles could only be partially achieved, partly due to objective obstacles that were mainly related to the reliability and limitations of the information provided in the analyzed studies. One can only assume that only a limited part of the published literature provides reliable information on the composition and structure of the materials used as adsorbents, not to mention a thorough characterization of all the relevant parameters. Very few studies address a deeper understanding and interpretation of results based on the applicable laws of physics that reflect physical reality, while analyzing a thorough characterization of an adsorbent’s structure and adsorption sites.

Two main questions have been raised in the studies dealing with the adsorption of dyes on clays and clay minerals. The first is to conduct an experiment and develop methods to reliably measure adsorption. This should not usually be that difficult, since the dye’s concentration can be easily measured in the visible part of the spectrum, even at very low concentrations. The second part is the interpretation of the results, which is much more problematic. In many cases, interpretations are based on incorrectly chosen theoretical models, as explained in the next chapters. An example is the use of the simplest adsorption models for very complex and multicomponent systems with adsorption centers of different types. Many interpretations are too simplified and neglect the influence of other phenomena and reactions, such as protolytic equilibria, dye molecule aggregation, solubility problems and dye precipitation, clay colloid destabilization, redox equilibria, and dye decomposition. Therefore, it is not easy to provide a relevant analysis on this topic that highlights all the critical issues that have been overlooked in the published studies.

Research papers on this topic can be divided into several groups, depending on their content and scientific objective: The simplest studies are reports that inform that a certain sample of clay or clay mineral, either in its original natural form or appropriately modified, can adsorb a certain organic dye. They usually do not provide a scientific interpretation nor have an ambition for a deeper analysis of the problem. Many such studies are related to some local practical problems, e.g., environmental pollution and the treatment of industrial wastes, and the physical interpretation of the results is considered irrelevant. The characterization of the adsorbents in these reports is often non-existent or inadequate, making a deeper understanding or interpretation of the data impossible.The second group of reports can be described as scientific studies that attempt to provide a deeper interpretation of the adsorption results, but in terms of scientific knowledge, they do not offer more than the previous group of studies. The main problem is often the lack of data on the composition and characterization of the adsorbents used. Adsorption has rarely been studied on purified and perfectly characterized monomineral samples without the influence of impurities that are optimally isolated from clay standards, which is well described in the literature. In most studies, admixtures and impurities may have had a significant effect on adsorption, leading to inadequate interpretations of adsorption data. Searching and selecting relevant and valuable papers from thousands of previously published reports is like looking for a needle in a haystack. For these reasons, a thorough analysis of the current literature is almost impossible unless all factors that contribute significantly to adsorption properties are evaluated, and all necessary information about the materials is available. Even if perfectly pure mineral samples are used, the problem of interpretation is not necessarily solved. A clay mineral of any type does not represent a substance with a precisely defined composition, structure, and properties, but rather a group of similar substances, yet often with very different properties within the group.

Below is one of many examples: the adsorption of methylene blue (MB) to kaolinite. MB is the dye that has probably been studied the most to date. Kaolinite is a non-expanding clay mineral with no permanent layer charge, and therefore does not show significant adsorption of cationic dyes. Montmorillonite or other smectites whose particles carry a negative layer charge are much better adsorbents for MB and can adsorb it to near cation exchange capacity (~1 mmol/g) [22]. However, it has also been reported that kaolinite or kaolin are efficient materials for MB adsorption [23], which may be due to the presence of admixtures. Many authors were unaware that the presence of impurities, such as smectites, in a kaolinite sample can significantly increase the amount of cationic dyes that are adsorbed. The detection of a smectite admixture in kaolinite samples is a problem [24]. Organic impurities, including those of biological origin, may have had a similar effect [25]. Higher adsorption values than expected were generally not confronted with an accurate analysis of the mineral composition of the adsorbent. No attempts were made to exclude the presence of interfering impurities via the isolation or thorough purification of the target mineral phase. I am not aware of any studies in which chemometric or statistical methods were applied to the multifactorial analysis of dye adsorption to determine the role of individual mineral components in clays, soils, or complex mixtures. In some studies, the authors did not even recognize the different meanings of the terms kaolin (rock) and kaolinite (mineral), and sometimes assigned kaolin to “raw” or “crude” kaolinite [26], or “natural” or “unpurified” kaolinite or clay mineral, or used terms which are contradictory by definition, such as “pure kaolin” [23]. Adsorption to materials that are a mixture of minerals may be misinterpreted if they are considered as pure substances. There are databases that contain several dozen papers that have addressed the adsorption of MB to kaolin or kaolinite, including modified materials in some studies. Some of these papers focus primarily on adsorption; in others, adsorption is only one of the research methods used to characterize materials. Some of the papers in this list have also extended adsorption to modified materials, other minerals, materials, or other organic dyes. The problems that have been summarized in this introduction are shown in the diagram in Figure 1. Not only is the importance of the purity of clay minerals underestimated, but the specific properties of clay minerals as adsorbents are often not considered. There are clay minerals of different types with different structures. However, even samples of minerals of the same type can vary in their composition to a greater or lesser extent, which can lead to differences in their properties. For example, the composition of the layers can significantly affect the layer charge, but also the properties of the broken bonds at the edges of the layers, such as the acid/base properties, the presence of groups that are able to act as ligands in coordination bonds, redox sites, and the like. Exchangeable cations are also very important, which, in the case of smectites, have a decisive influence on swelling properties. The size of the particles, together with the layer charge, is one of the crucial parameters affecting the stability of colloidal systems. It affects the access of molecules to the surface of clay particles. Many parameters of clay minerals and clays influence each other or are closely related. The modification of clays often leads to qualitative changes that increase the complexity of the subject.

The above facts were considered in formulating the objectives of this review. Instead of an in-depth analysis of the huge number of publications that deal with the adsorption of dyes on clays and clay minerals, some of them of low value, this review aimed to critically analyze the phenomenon of the adsorption of organic dyes in terms of the fundamental laws of physical chemistry, taking into account the structural, physical, and chemical complexity of clays and clay minerals, and also considering the phenomena that accompany or are part of adsorption, and all the problems that complicate the interpretation and understanding the phenomenon. It is critical for the methodologies used and in searching for new directions. The types of modified clay minerals and complex materials, and their significance for industrial applications, are only briefly overviewed.

## 2. Theoretical Basics

Adsorption leads to an increase in the local concentration of the adsorbed molecules of a substance, called a sorbate, on the surface of an adsorbent. Adsorption is a thermodynamic quantity that is defined at equilibrium for a given system, and depends on the temperature and concentration of the substance. Most studies have focused on the adsorption of organic molecules using basic adsorption models that are based on the fundamental theories of Langmuir or Gibbs. Both models were originally developed for adsorption from a gas phase onto adsorbing solids. Both theories are basic parts of general and physical chemistry monographs and chapters related to adsorption.

The Langmuir theory of adsorption is based on a kinetic model in which adsorption and desorption occur simultaneously, and the adsorption and desorption rates and the number of free adsorption sites are the main parameters affecting the process [27,28]. The adsorption rate depends on the number of active interactions with the surface that is not yet covered by the adsorbent molecules. The desorption rate depends on overcoming the activation energy that is associated with the adsorption enthalpy. The validity of the Langmuir model is limited by assumptions that apply primarily to ideal cases. For example, the surface area of the adsorbent is considered to be homogeneous, concerning the adsorption energy. The molecules in the adsorbed state do not affect each other and there is no migration of molecules on the surface. Each adsorption site can bind only one molecule. Nevertheless, the adsorption of cationic dyes on Na^+^-smectites was often described using the Langmuir isotherm, suggesting that the cations adsorb homogenously and do not interact with each other [29]. The complexness of the adsorption process in dye/clay systems was ignored in most studies that apply this model, as has been critically reviewed in some studies [30,31].

The second most commonly used model is based on the Freundlich adsorption isotherm [32,33]. It was originally proposed based on empirical observations and is not applicable at high concentrations near the saturation point of an adsorbent surface. Gibbs’ view from the thermodynamic standpoint led to the derivation of other isotherms. The simplest type, which also takes into account the lateral interaction between adsorbed molecules, is the Fowler–Guggenheim model [34,35]. The model can be uniquely described by the change in isosteric heat as adsorption increases. The isosteric heat represents the thermal effect, resulting from an infinitesimally small increase in adsorption. When positive interactions occur between adsorbed molecules, there is a positive contribution of isosteric heat. In the case of repulsive forces, the coefficient has a negative sign. If the interactions between molecules are negligible, a zero change is expected.

Contrary to the models that were originally developed for adsorption from a gas phase, the adsorption of organic dyes from a liquid phase is a more complex phenomenon. The simplest systems are represented by dilute solutions of non-electrolytes interacting with the surface of an adsorbent. Even in this simple system, another component to deal with is the solvent molecules. The process of the adsorption of dye molecules on a solid surface (s) such as clay mineral particles can be described using the following equation:dye(aq) + H_2_O(s) ⇌ dye(s) + H_2_O(aq)(1)

The label (s) indicates the adsorbed state of dye or water molecules on a solid; (aq) indicates the presence of a molecule in the bulk of an aqueous phase. The general thermodynamic equilibrium constant can be defined as follows:(2)$$K=\frac{x_{d}a_{w}}{x_{w}a_{d}}$$
where *a*_w_ and *a*_d_ are the activities of the solvent and dye in a liquid phase. The activities in the adsorbed layer on a solid surface are expressed as mole fractions, *x*. The symbols d and w stand for dye and water, respectively. For dilute solutions, *a*_w_ = 1, and the ratio *K*/*a*_w_ represents a constant *b*. The mole fraction *x*_d_ can be defined as:(3)$$x_{d}=\frac{n_{d,s}}{n_{s}}=\theta =\frac{ba_{d}}{1+ba_{d}}\approx \frac{bc_{d}}{1+bc_{d}}$$

Here, *n*_d,s_ and *n*_s_ are the amounts of adsorption sites per unit mass of adsorbent (expressed, for example, in mol g^−1^). They refer to the sites occupied by the dye molecules (*n*_d,s_) vs. the total number of adsorption sites (*n*_s_). The activity, *a*_d_, can be expressed as the concentration (*c*_d_) in dilute systems. *θ* stands for the fractional occupancy of the adsorption sites. 

A solvent can affect solute adsorption through the interaction of the solvent molecules with the surface of the adsorbent, solute–solvent interactions, and more importantly, through the interaction between the solute molecules themselves (Figure 2). Molecular aggregation is a typical example [36,37,38]. Entropic factors in the desorption of a number of small solvent molecules exchanged by a relatively larger solute molecule may also play a role [37,39]. Langmuir’s isotherm has often been used as a simplified model for the adsorption of dyes from dilute aqueous solutions. In contrast to gas adsorption, competitive adsorption between solvent and solute molecules occurs in the aqueous systems of dyes and clay minerals. Therefore, it is very important to be careful when defining the adsorption model in such systems. When applying the simplified model, the differences between an ideal and a real system must be taken into account, and the effects resulting from these differences must be estimated. For example, the contributions of the interactions of the adsorbed dye molecules with solvent molecules and between solvent molecules themselves are usually considered to be canceled when the Langmuir model is applied. 

## 3. Adsorption vs. Ion Exchange Reaction

Organic dyes represent a less polar component in their aqueous solutions and are therefore preferentially adsorbed onto less polar surfaces or adsorption sites. The hydrophobic effect contributes significantly to the adsorption of dyes from aqueous solutions, e.g., [10,37,40]. The hydrophobic interactions between adsorbed nonpolar molecules or their groups are secondary contributors to the enhancement of adsorption. In other cases, Coulomb interactions may contribute, especially for organic dyes with ionic groups and adsorbents with charged surfaces. The adsorption of ionic dyes on the surface of minerals with an opposite charge can be very strong and irreversible [41,42,43,44]. Therefore, the mechanisms of the adsorption of ionic dyes deserve special attention. One of the most common examples is the ion exchange reaction, leading to the replacement of Na^+^ ions with dye cations, which occurs in the colloidal systems of Na^+^-smectites. The general equation for ion exchange can be defined as follows:dye^+^(aq) + Na^+^(s) ⇌ dye^+^(s) + Na^+^(aq) + (4)

In electrolyte solutions, the activities are directly proportional to the concentrations, and the thermodynamic equilibrium constant is expressed using the activity coefficients, *γ,* and the molar selectivity coefficient or the concentration equilibrium constant, $K_{{dye}^{+},{Na}^{+}}^{c}$:(5)$$K_{{dye}^{+},{Na}^{+}}=\frac{a_{{Na}^{+}(aq)}a_{{dye}^{+}(s)}}{a_{{dye}^{+}(aq)}a_{{Na}^{+}(s)}}={K_{{dye}^{+},{Na}^{+}}^{c}\frac{\gamma_{{Na}^{+}(aq)}\gamma_{{dye}^{+}(s)}}{\gamma_{{dye}^{+}(aq)}\gamma_{{Na}^{+}(s)}}}_{}$$

The selectivity coefficient can be expressed as
$$K_{{dye}^{+},{Na}^{+}}^{c}=\frac{c_{{Na}^{+}(aq)}c_{{dye}^{+}(s)}}{c_{{dye}^{+}(aq)}c_{{Na}^{+}(s)}}$$

In general, the adsorption of organic cations on the surfaces of smectites is strong and effectively leads to the exchange of inorganic cations for organic ones [45]. This is consistent with the nature of organic cations being soft acids, i.e., molecules with a delocalized positive charge [46] that preferentially form strong bonds with soft bases, as represented by surfaces with a delocalized negative charge, as on the surface of smectites [15]. However, the strong adsorption of organic cations should be seen from a much broader context. The irreversible adsorption of organic substances [47] has often been observed, as mentioned above. One of the typical examples is the adsorption of polymeric substances, but it also occurs with smaller molecules. The cause of irreversible adsorption may be due to kinetic factors. Adsorbed molecules alter their properties by forming various associates called molecular aggregates [13,37,42,48,49,50,51,52], which form condensed monomolecular layers between clay particles. These forms are stabilized by the interactions between the molecules, which, in this state, exhibit a very slow desorption rate. It should be noted that, even in this case, secondary desorption can be achieved by changing the solvent. Desorption is also possible when a solute that exhibits an even higher affinity for the sorbent surface is added to the system [53,54].

## 4. Adsorption Mechanism of Organic Dyes on Clay Particles

### 4.1. Basic Concepts

In terms of kinetics, the adsorption and accompanying processes can proceed at different rates, depending on the availability and surface activity of the clay mineral particles and the type of forces between the dye molecules and the mineral surface that are used in the adsorption. Rapid, diffusion-controlled near-instantaneous adsorption processes are applied in some cases of strong interactions between the dye cations and the freely accessible, negatively charged mineral surface [55,56,57,58]. The condition is the state of the colloidal system in which the particles have a sufficient surface area and, in particular, a sufficient capacity to adsorb the amount of dye present. In case the clay mineral particles form agglomerates, access to a part of the surface of the particles inside the body of the agglomerate may be limited [59,60,61,62]. Adsorption on clay particle agglomerates is realized as two types of processes to two types of non-equivalent surfaces. The first one is represented by a fraction of the molecules adsorbing very rapidly onto a particle surface that is easily accessible, and the second is via the rest of the molecules migrating inside the agglomerate and proceeding more slowly. The structure of the agglomerate, the arrangement of the particles in the agglomerate, swelling, and the interlayer distances or even the size of the individual particles forming the agglomerate are all important factors that may influence the slower processes that are described in more detail in next chapters.

### 4.2. Initial Adsorption

The adsorption kinetics of the very fast process is relatively difficult to characterize, as common experimental methods that are fast enough to record the changes in the state and properties of dye molecules in such a fast adsorption process are not available. Small changes in the spectral properties of the adsorbed molecules compared to the free molecules in a solution can be quite a common phenomenon that is sometimes used to monitor fast adsorption processes [37,51,52,63,64,65,66,67]. Dye adsorption often leads to molecular aggregation. Dye molecular aggregates are species that exhibit significant changes in the properties of the chromophores, and, therefore, can be easily detected and distinguished from free, non-adsorbed molecules [36,37,58,68,69]. However, in many cases, adsorbed dye molecules are initially adsorbed as non-aggregated species, and the formation of aggregates takes place over time during slower processes [58,70]. There are only a few studies in which an instantaneous process of dye adsorption has been analyzed in more detail [56,58]. However, more accurate studies that provide the kinetic parameters for rapid initial adsorption are still missing [71,72,73,74,75]. The whole rapid process can also be divided into two stages, faster and slower. The first step is the diffusion of molecules through the bulk liquid toward the surface of the particles. Diffusion-limited reactions are extremely fast in mixed reaction systems with rate constants in the range of 10^9^ mol^−1^ L s^−1^, depending on the viscosity of the medium. The second, slower stage is the adsorption on the surface of the adsorbent particles. In this case, the solvent molecules that were originally adsorbed on the surface of the sorbent are exchanged for dye molecules [76,77]. In the case of ion exchange, the exchangeable cations or their aqua complexes are exchanged for dye cations. The bonds between the surfaces of the mineral particles and the original solvent molecules or the original cations being exchanged exhibit some energy barriers in this process, which may be reflected in the slowing of the exchange process and the kinetics of the overall adsorption. However, in many cases, this is not the case. For example, when the parent cations do not bind directly to the surface of the particles but are present as aqua complexes in a diffuse electrical double layer at a distance of a few nm from the surface of the particles, the electrostatic attractive forces between the cations and the surface are relatively weak and the exchange occurs easily and relatively rapidly. This is evident in stable dispersions of Li^+^- and Na^+^-forms of smectites, where the thickness of the electrical double layer can reach values of up to tens of nm [78,79,80,81]. On the other hand, some of the inorganic cations form relatively strong and direct electrostatic bonds with the surface and bind directly [82,83], for example, via interactions with oxygen atoms, forming pseudo-hexagonal cavities on the surface of smectite particles (K^+^, Rb^+^, Cs^+^), and possibly also via hydrogen bonding (NH_4_^+^). This is evidenced in some phenomena related to the formation of molecular aggregates in the colloids of K^+^-, NH_4_^+^-, Rb^+^-, and Cs^+^-montmorillonite [80,81]. The exchange of such cations may be slower with a higher energy barrier. Moreover, in these and other cationic forms, the lower stability of the colloidal systems often affects the whole process. It plays a role in particular when an electrolyte contains divalent and trivalent cations, which reduces the thickness of the electrical double layer and significantly destabilizes colloidal systems [79,84].

In summary, the initial adsorption process can be much faster and finish before completely mixing the liquid phases of the two components. This is the case when the adsorption kinetics are on the time scale of in-solvent-diffusion-controlled processes or when the exchange of molecules on the surface has no appreciable energy barrier. In such a case, the process of rapid adsorption may initially lead to a heterogeneous distribution of dye molecules on the surface of the particles. The result is the presence of molecules on the surface of the particles that the molecules first come into contact with. The rest of the particles may be without adsorbed molecules, or some may be only partially covered with adsorbed molecules. The evidence is a very fast process of the molecular aggregation of some types of dyes [42,61,81,85,86,87,88].

### 4.3. Slower Processes following the Initial Adsorption

It follows that slow processes are related to the reorganization and redistribution of initially adsorbed molecules. It must proceed via the desorption of adsorbed molecules. Desorption is one of the foundations of the theoretical model of the Langmuir isotherm. Desorption can lead to the fast re-adsorption of molecules at the same or a nearby site [89,90,91,92,93]. The release of molecules from the surface of the particle is also possible, allowing for the transport of the sorbate molecule from one particle to another [94,95,96]. Slow processes can lead to a change from dense local adsorption and the heterogeneous distribution of molecules on a fraction of the total surface, to a more homogeneous distribution of more uniformly distributed molecules on the surface of all particles. From this point of view, the redistribution of the dye molecules can be understood as a spontaneous process with an increase in entropy. From the point of view of kinetics, two independent steps can be distinguished: these include the desorption process itself and the subsequent diffusion of a molecule to a new adsorption site. Overcoming the bond between the surface and the molecule represents a certain energy barrier, *E*_d_. The kinetics of this process can be described with a simplified relation of the Arrhenius law:(6)$$k_{d}=Ae^{-\frac{E_{d}}{RT}}$$

*k*_d_ is the rate constant of the process, *R* is the universal gas constant, and *A* is the pre-exponential factor. The desorption half-life, $\tau_{1/2}$, can be derived from the activation energy and the temperature, *T*:(7)$$\tau_{1/2}=\tau_{0}e^{\frac{E_{d}}{RT}}$$

The high adsorption energy that is expected when cationic dyes are adsorbed on the surface of smectite particles can be a major obstacle and lead to a slowdown of the desorption process [37,52,58,97].

As mentioned earlier, a released molecule can change adsorption sites via diffusion. The theoretically simplest model would be defined as a diffusion process on the 2D surface of a single clay mineral particle. This can occur considering stable colloidal systems (such as dilute Li^+^- and Na^+^-smectites, low electrolyte concentrations, and low concentrations of particles and dye) where single particles are present and this state does not change after the dye adsorption. Diffusion at the surface of a single particle is a process that depends strongly on temperature to overcome a certain energy barrier of attractive forces between the dye molecule/cation and the surface:(8)$$D(T)=D_{0}e^{-\frac{E_{D}}{RT}}$$

Here, *D*(*T*) is the temperature-dependent diffusion coefficient and *D*_0_ is a constant. The energy barrier is estimated to be slightly smaller than the value of the desorption activation energy, *E*_D_. It varies considerably in the nanometer range on the surface of the adsorbent, and its distribution is related to the distribution of adsorption sites with an increased affinity for the diffusing dye molecule. In any case, surface diffusion is a more complicated process than simple Brownian motion [91]. It is not entirely clear what role is played by collisions between the adsorbed molecules on the surface and the solvent molecules from the bulk, or whether collisions between particles play a role in the activating processes. Theoretical models of these processes have already been supplemented by experiments proving that the surface diffusion of adsorbate molecules is an interesting and relatively complex phenomenon [91,93,98,99,100]. The trajectories of molecules during diffusion are composed of several phases, from the slow migration of molecules on the surface, to jumping from one surface location to another, to diffusion in the bulk phase over relatively long distances near the particle surface [93]. Moreover, this type of diffusion has some interesting properties that one would not expect in a random process of 2D Brownian diffusion. During this process, there is a certain type of selection for adsorption sites over time. In the phase of fast adsorption, the adsorbed molecules occupy random centers on the surface of the particles that they encounter first; in slow diffusion, the molecules preferentially terminate diffusion at sites that stabilize the final state, for example, at adsorption sites with a strong binding of the molecules to the substrate [91,93]. In this way, selective adsorption can be applied when there are no equivalent adsorption centers on the surface of the particles. As far as I know, this model has been confirmed by a limited number of experiments. On the other hand, it was recently confirmed to take place in the phenomenon of the molecular aggregation of an organic dye on smectite particles [58,73] (Figure 3). Over time, the number of molecular aggregates increased and reached limiting values, but the fraction of aggregated molecules was strictly limited by the presence and number of suitable sites promoting aggregation [73]. It is interesting, and at first glance contradictory, that the diffusion processes, which led to a redistribution of molecules and thus also reduced the local surface concentration, led to an increase in molecular aggregation. A higher degree of aggregation requires a higher local concentration, which at first glance is a contradiction. It can be explained that the redistribution of molecules allows them to reach sites on the surface that support aggregation. In clay minerals, the presence of non-equivalent centers for the adsorption of molecules is to be expected, and of the most diverse types. Examples include sites with a negative charge in the layers caused by isomorphous substitutions in the tetrahedral and octahedral sheets, sites on the basal surface compared to the edges of the layers, sites with domains of different compositions, the heterogeneous distribution of charge centers, etc.

In addition to on-particle or lateral diffusion, which is strictly limited to the movement of molecules on a single particle, the migration of molecules between two particles has also been observed and the mechanism is supported by experiments [73]. Such a phenomenon is called interparticle diffusion, and can occur when the molecule can leave the surface of the particle along a trajectory toward the liquid phase. The free molecule in such a state migrates in the solvent and can reach the surface of the second particle in its vicinity. For diffusion in a liquid medium, the relationship was introduced independently by Einstein and Smoluchowski:(9)$$D\propto \frac{k_{B}T}{\eta r}$$

The diffusion coefficient of a particle in the liquid phase (*D*) is a function of temperature (*T*), the dynamic viscosity of the medium (*η*), and the radius of the molecule/particle (*r*). *k*_B_ is the Boltzmann constant. Assuming that the viscosity of the medium is similar to that of aqueous solutions, the distance traveled per unit of time is greater, and thus diffusion is faster than the processes occurring at the surface of the adsorbent. In summary, the diffusion processes that take place after the desorption of the molecules can be divided into slower (surface or lateral diffusion) and faster (diffusion in the liquid phase) phases.

The same processes and mechanisms are believed to apply not only to dyes [73], but also to other organic/inorganic molecules and ions. Slow diffusion processes can be of great importance in adsorption related to chemical catalysis, for example, in the degradation of organic pollutants [101,102]. The initial adsorption of organic pollutant molecules may not lead to effective catalysis of their degradation for a simple reason: the surface of the particles may contain only a relatively small number of catalytically active centers. The interaction of the initially adsorbed molecules with these centers is unlikely and a large fraction of the molecules may be adsorbed on inactive sites. However, the diffusion processes may allow for the redistribution of the molecules on the surface of the particle or between particles, increasing the probability of their interaction with the catalytic centers and promoting the gradual degradation of the pollutant [103,104]. However, methods for the sensitive identification of adsorbed forms or molecular aggregates based on spectroscopies in the visible region that can clearly distinguish them from non-adsorbed molecules exist only for dyes [51,52].

### 4.4. Rapid and Slow Diffusion as Parts of the Complex Process

The problem with capturing diffusion processes is that they occur in the second stage as a subsequent process after the relatively rapid initial adsorption on the available surface of the adsorbent. The diffusing molecules leave the original adsorption site but find new ones, resulting in essentially no change in the adsorbed amount of dye molecules, which therefore cannot be recorded as a change in that amount. The rate of initial adsorption is very high and the progress of the process is not detectable via conventional methods. An example is the adsorption of cationic dyes on the surface of smectites when the amount added is far below the value of the cation exchange capacity, usually at a few percent of the CEC. Often, all of the added dye is adsorbed immediately, as shown through the absence of dye in the supernatant, separated via filtration or ultracentrifugation, and also via spectral methods. An important sign of complete adsorption is often significant changes in the spectral properties, such as shifts in the absorption bands and changes in luminescence. Later changes in spectral properties are often the result of slower diffusion processes that follow compared to the initial adsorption and lead, for example, to molecular aggregation [37]. The formation of molecular aggregates often occurs much more slowly than the initial adsorption of dyes, and changes in spectra can be observed in seconds, minutes, or hours. The sites of initial adsorption may not lead to molecular aggregation [58,105,106], or they may lead to the formation of aggregates that are different from those that form over time due to diffusion processes [107,108]. If there were no methods for the very detailed spectral characterization of the molecular aggregates of organic dyes, it would not be possible to follow these slow processes or to describe their mechanisms [88]. The identity of slower lateral or intraparticle diffusion and faster interparticle diffusion in the same experiments was identified in the paper that dealt with dye molecular aggregation [73] (Figure 4).

## 5. Dye Adsorption and Destabilization of Colloidal Dispersions

Clay minerals that have particles with surfaces carrying electrostatic charge sites effectively adsorb ions of the opposite charge. Adsorption to low extents does not have to lead to significant changes in the electric potential at the adsorbent surface. An example is smectites, whose surfaces have a permanent negative charge, the origin of which is related to the presence of non-equivalent isomorphic substitutions in the layered particles. The permanent charge persists under a broad range of conditions and is independent of pH. However, if the adsorption degree of ions is significant and leads to the formation of ion pairs, or the adsorption takes place in parallel with a protolytic reaction between the surface and the dye or solvent molecules, the potential on the Stern layer of the adsorbent decreases and can be completely neutralized [72,109,110,111,112]. The reduction in the electrostatic potential reduces the repulsive forces between the particles, leading to the destabilization of a colloidal system. If electrostatic repulsion was a major factor in the stabilization of the colloidal particles, the adsorption of oppositely charged molecules could lead to particle flocculation [72]. However, if the adsorption of dye ions continues beyond the point of zero charge value, then the charge on the Stern layer can be reversed. The particles obtain a charge of the sign of the adsorbed ions, which may re-stabilize the colloidal dispersion again [72,113,114,115,116]. Dye molecules, including ionic dyes, are more hydrophobic than inorganic ions. Therefore, the exchange of inorganic cations for organic ones causes the surface to become less hydrophilic with increasing amounts of the adsorbed organic molecules of a lower hydration energy [117,118,119,120]. The hydrophobic surface can bring about the reduction of the surface energy between particles and water molecules, which is another factor that contributes to the destabilization of the colloidal systems [72]. Destabilization mostly comes from the changes occurring at the basal surface, which is the dominant surface fraction. However, it can also be related to the interactions occurring at the particle edges. For example, stable aminopropyl functionalized magnesium phyllosilicate was applied as an adsorbent for the textile anionic dye Reactive Red 120. Efficient adsorption led to the precipitation of the dye/silicate complex [121].

In summary, the destabilization of colloidal systems leads to the formation of particle agglomerates, which are relatively complex systems. The processes that occur after initial adsorption, such as desorption, diffusion, and reabsorption, also occur in agglomerates but proceed via more complex mechanisms. The formation of particle agglomerates should be considered as being a part of the models that describe dye adsorption on clay surfaces.

## 6. Dye Adsorption on Clay Particle Agglomerates

The above analysis of diffusion processes is based on the simplest system and assumes the presence of individual particles and a sufficient surface area being freely available for the adsorption of molecules. However, single particles are expected only in dilute colloids of smectites with a relatively low layer charge and a small particle size, and mainly in systems with low ionic strength (Li^+^-and Na^+^-forms). Very often, the colloidal system of adsorbents is formed by agglomerates of layered particles, e.g., [49,122]. In such a case, the most accessible surface is the outer surface of the agglomerates, which may include the basal surface, but also the edges of the particles. In such a case, the mechanisms of the diffusion processes may have a much more complex character. The colloidal system may not have enough free, easily accessible adsorption sites for the initial adsorption of the entire amount of dye molecules. In other words, agglomeration leaves much of the surface area inside the agglomerate in the form of interlayer spaces that are still accessible to sorbate molecules, but with a higher barrier than the outer surface. The reduction in the freely accessible outer surface area can be considerable for large agglomerates of particles. If the edges of the particles are neglected, an agglomerate consisting of a number of particles of the same size (*N*) has the same external surface (*S*_aggl_) as a single, layered particle (*S*_part_). The internal surface area of such an agglomerate can be expressed using (*N*−1)*S*_part_. The fraction of the external free surface area to the total surface area can be expressed as follows: *S*_part_*/NS*_part_ = 1/*N.* A slightly more complicated estimate would be to account for the fact that the molecule in the interlayer space interacts with the basal surfaces of the two adjacent particles, which would reduce both the internal and total surface areas compared to systems based on single particles. In any case, it can be summarized that the formation of agglomerates of particles leads to a significant reduction in the readily available surface area. Therefore, the adsorption of molecules is likely to occur most rapidly at the freely accessible outer surface and may be significantly slower in the case of intercalation between layers, i.e., into the inner surface of the particle agglomerate. In addition, the kinetics and degree of adsorption at the inner surface can be influenced by various factors, such as the extent of agglomeration (number of agglomerated particles, *N*), interlayer expansion, the type of exchangeable cations, the cation exchange capacity, the type of dye, the modification of the adsorbent, etc. These parameters are often closely related. For example, intercalation into the interior of tactoids with a relatively large interparticle expansion may occur without a significant barrier and comparatively rapidly, as at the outer surface. However, if the expansion of the interlayer spaces is small and there is also a large amount of tightly bound inorganic cations in the interlayer spaces, the intercalation of dye molecules may be associated with a high energy barrier.

Assume that adsorption is faster on the outer surface and a larger fraction of the dye molecules are preferentially adsorbed there. Even a very small number of organic molecules can significantly saturate the adsorption sites on the outer surface of the agglomerate, while the spaces between the layers remain free from the point of view of adsorption. A similar situation occurs with non-expandable clay minerals such as kaolinite, pyrophyllite, illite, and mica. The adsorption of molecules/ions, leading to the saturation of the outer surface, can significantly affect the properties of the surface. For example, the replacement of inorganic cations with organic cations leads to a significant hydrophobicity of the surface. Such a surface may become unstable in an aqueous environment and tend to combine with another layered particle to increase the size of the agglomerate. Agglomerates of particles can be considered as dynamic systems [123] in which there is a continuous disintegration of agglomerates into smaller units and, conversely, the formation of larger agglomerates by merging from smaller units. When part of the outer surface is saturated with nonpolar organic molecules, association with another particle or aggregate is likely and the original outer surface becomes an inner surface with a relatively high degree of intercalation of dye molecules (Figure 5).

In principle, a similar phenomenon can occur in adsorption on single-layer particles (Figure 5b). Since the size of the freely available surface area is much larger in this case, a much larger number of dye molecules is required to destabilize the colloidal system [124,125,126,127]. As a result of dye adsorption, individual particles in originally stable colloidal systems transform into small agglomerates with two or more stacked layers, which can gradually lead to a higher degree of association, as well as the partial flocculation and destabilization of the colloidal systems. An interesting phenomenon is the formation of stacked bilayers in colloids of synthetic hectorites with large particles [128,129,130]. In this case, the preferential formation of a phase of intercalated molecules between two hectorite particles was observed, while the outer surface of the stacked bilayers remained unchanged. The mechanism of the formation of such systems can be understood based on the initial adsorption on the surface of a layered particle, the hydrophobization of this surface promoting the adsorption of further molecules, and the subsequent association with another particle (Figure 5b), but other mechanisms can be further designed. A phase of intercalated hydrophobic molecules forms between particles that are associated in this way, which can be thermodynamically stable and prevents the redistribution of intercalated molecules on the outer surface [131]. In other words, the stability of the phase containing the intercalated molecules may prevent them from being desorbed via diffusion processes and migrating to the surface of other particles. As already shown, the phenomenon does not need to be restricted to the organic phase [132]. A similar phenomenon was observed in the molecular aggregation of rhodamine dye on the surface of saponite particles [97]. An increase in the ratio of dye to saponite resulted in an increased amount of aggregates formed in the initial phase, but also in an increased probability of particle agglomerate formation. The agglomeration of particles led to the entrapment of molecules between the layers, which prevented the redistribution of cations and, consequently, significantly reduced the formation of aggregates formed over time through the diffusion of the dye molecules [97]. In principle, a similar phenomenon is the fixation of the molecules in the channels of fibrous clay minerals. Palygorskite exhibits a high affinity for the irreversible adsorption of some organic dye molecules, which prevents the regeneration of the adsorbent. The phenomenon was explained by the formation of stable dye/silicate complexes, similar to Maya blue [133]. The scheme in Figure 6 summarizes all the main phenomena that may follow the initial adsorption of organic dye molecules on clay mineral particles.

The intercalation of organic dye molecules between clay layers inside the existing particle agglomerates can become very important in some cases, accounting for most of the total adsorption. This occurs particularly at higher ratios of dye to clay mineral and in systems with a high formation of particle agglomerates, when the outer surface does not have the capacity to bind the added amount of dye molecules. The kinetics of such a process can be significantly affected by the interlayer expansion, which depends on the type of clay mineral, the charge of the layers, the type of exchangeable cations, the size of the particles, and the reaction conditions. Intercalation often occurs only at sites between layers near the outer surface of the particle agglomerates. The fixation of molecules between the layers can block the entry of additional molecules into the interlayer spaces. An example of this is the negative effect of increasing the concentration of MB solution on reducing the adsorption capacity of fibrous clays [134]. The highest adsorption values were obtained only when a highly dilute dye solution was used. Apparently, the effective penetration of molecules into the structures of such materials requires a gradual process. For this reason, the kinetics of the intercalation processes may be significantly slowed down and depend on other processes and different types of interactions, such as the reorganization of particle agglomerates, slow diffusion processes within the agglomerate [134], etc. Expansion between layers always logically plays an important role, at least in the initial stages of intercalation. For example, kaolinite, as a non-expanding mineral, cannot accommodate significant numbers of organic molecules until the interlayer spaces have expanded with the help of some substances [135]. Since even this property is often lost via the de-intercalation of the modifier, kaolinite becomes an effective adsorbent by grafting the interlayer spaces or introducing charged groups between the layers [136]. In addition to increasing the expansion of the interlayer spaces, the specific interactions of the particular dye can also play an important role. An example of this is a comparison of the adsorption of alizarinate on Na^+^- and Al^3+^-montmorillonite [137]. One would expect that the expansion of Na^+^-montmorillonite would allow for better adsorption of the dye molecules, while the closed interlayers that are typical of the Al^3+^-form would be an obstacle to the effective penetration of the molecules. The exact opposite was observed when a 5-fold amount of the dye was adsorbed on the Al^3+^-form compared to the Na^+^-form. This is due to the formation of the Al^3+^ complex with alizarinate, which was the driving force for the penetration of the molecules between the montmorillonite layers. In contrast, alizarinate in the Na^+^-form formed complexes only with the aluminum atoms that were present on the outer surface at the broken bonds at the edges of the layers. Spontaneous reactions in the interlayer generally promote the intercalation of molecules. This was exploited in the intercalation of a laser dye in the form of a reactive silane into the interlayers of expanded kaolinite [138]. The ability of many molecules, even large ones, to penetrate between the layers of clay minerals is also confirmed by so-called solid-state intercalations. Such reactions take place without a solvent from a dry mixture of clay and organic matter (see citations in [139]). In such reactions, the solvent is present only in very small amounts, which precludes the effective expansion of the mineral. Nevertheless, it has been possible to intercalate a large number of organic dyes in this way. The formation of complexes with interlayer cations was also demonstrated here. An example is the intercalation of 8-hydroxyquinoline in Li^+^-, Zn^2+^-, and Mn^2+^-montmorillonite in the solid state [140]. A large number of examples of solid-state reactions have been analyzed in detail in a review article [141]. Many of the papers that were analyzed in this review also address the intercalation of dye molecules. The intercalation of dye molecules between layers of clay minerals is possible and in many cases very effective, although the extent of the process can be limited or its rate slowed down. As mentioned above, the course of intercalation is influenced by many factors, and the expansion between layers is only one of many. An accurate physical description of this process, including the processes that follow the initial penetration of the molecules, such as the diffusion of the intercalated molecules in the solid phase of the particle cluster, is not a simple task. Only descriptive models are possible, using functional dependencies that mathematically fit the course of intercalation. It is practically impossible to accurately describe such complex systems by considering all physical interactions and describe them quantitatively using simple models.

Clay minerals forming particle agglomerates in their colloidal dispersions represent relatively complex systems. A correct characterization of such systems is necessary for the interpretation of dye adsorption. Moreover, it is not possible to generalize, and the behavior of dye molecules in such systems depends strongly on the nature of the interaction between the sorbent and the sorbate molecules, the nature of the sorbate, the reaction environment, and many other parameters.

## 7. Critical Assessment of the Adsorption Models

Several theoretical models describing the mechanism of the adsorption of organic molecules have been proposed [2,3,5,6,9,10,14,142]. Probably, almost all of them were also used in the attempt to describe the adsorption of organic dyes on clay mineral particles. Very often, some serious contradictions between the theoretical assumptions and the physical state of the systems were overlooked, as has been critically evaluated in several articles [142,143,144,145,146]. Many of the models are based on the assumption of relatively simple systems with a uniform type of adsorption centers. Such models are commonly and incorrectly used for complex systems of modified clays, ternary and multicomponent systems, and composites. The theoretical basis of the models, based on simple assumptions, limits their applicability, especially for more complex systems. On the other hand, the fit of measured data to the function defined in the model is chosen as the most frequent criterion to verify the applicability of a model. The most commonly used method is regression analysis, which evaluates how well the measured values match the assumed functional dependence of the given model. The correlation coefficient or other statistical parameters are considered as the main criteria of whether a given model can be applied to a system or not [147]. At the same time, the theoretical basis of the model often directly contradicts the model selection criteria, especially in the case of complex systems. A high correlation coefficient is often a matter of chance when the mathematical function describes relatively well the changes in adsorption with time or the changes observed with increasing equilibrium concentrations. Only a small fraction of papers have been critical of these problems, and such critical papers are still in the minority [142,143,144,145,146]. Data calculated directly from the functional dependence of one model may yield relatively high correlation coefficients with another model [145]. Therefore, a purely mathematical approach, since it is also most commonly used, only adds chaos to the whole matter. Therefore, the mathematical model should be primarily selected based on correct theoretical assumptions that reflect the reality of the studied system, and only secondarily verified by fitting the measured values.

The most contrasting example is adsorption kinetics, according to the pseudo-second-order model, and in particular, its use in the form of a linearized function [143,144,145,146,148]. In the linearized model, time as a physical quantity appears as part of the dependent and independent variables. Since the dependence of time is formally linear to time, it is sufficient if the effects of the other parameters are not significant, and perfect linearity is achieved. False ideal linearity occurs mainly in cases where adsorption approaches the equilibrium state and does not change much with time or increases only slightly significantly. On this basis, thousands of papers emerged with a wrong interpretation of adsorption kinetics, attributed to the mechanism of the pseudo-second order model.

The linearization of originally nonlinear models is generally a problem because the nonlinear treatment distorts the distribution of experimental errors [33,146,148,149,150]. One of the possible approaches to avoid this is to use nonlinear regression, where directly measured data or their linearly treated values appear as variables. An example of this is the dependence of adsorption values on equilibrium concentrations, while both quantities could also be recorded as measured values of dye absorbance via spectrophotometry, since both quantities are in a linear functional relationship, resulting from the Lambert–Beer law. Similarly, for kinetics, nonlinear regression should be applied to nonlinear models where adsorption and time appear directly as dependent and independent variables, respectively.

However, even this approach does not guarantee a successful outcome of the analysis of the theoretical model and the correct identification of an adsorption mechanism. As mentioned above, there is a risk that one data set will give a good nonlinear regression result for several mathematical models. Moreover, in hundreds of publications, the analyses have been performed only with a limited number of measured values, which were often subject to large experimental errors. Even with a well-designed analysis, it is not possible to obtain reliable results in such cases. The number of measured adsorption points for adsorption isotherms or kinetics is rarely more than 10. Very complex models describe complex systems where several different types of adsorption centers are expected. Such models would require a large number of measured values that are not subject to large experimental errors. It is rather rare when tens to hundreds of measured points are used in the model analysis [58,73,97].

Other problems are related to experimental accuracy [151]. The determination of the correct amount of adsorbed dyes in clay mineral systems often encounters specific problems. One example is the separation of the supernatant, which contains the non-adsorbed portion of the substance, from the clay mineral particles. When the clay mineral consists of particles of small size, all common separation methods such as decantation, filtration, and centrifugation are problematic or inefficient. A typical example is the commercially available synthetic trioctahedral smectites saponite and hectorite, which form stable colloids based on very small particles. Even if the two phases of clay and supernatant are successfully separated, the equilibrium may be disturbed by the treatment method and the actual amount of the adsorbed substance may change during the application of the separation process. A particular problem is the measurement of adsorption kinetics, especially in cases where the process is relatively fast and the amount of the adsorbed substance may change with relatively slow sample processing. In such a case, the prospects for characterizing slow adsorption processes are much better.

On the other hand, the adsorption of dyes on clay mineral particles also offers considerable advantages from an experimental point of view, which has already been briefly mentioned above. The dye very often changes its photophysical properties, which can be detected spectrophotometrically [37,48,58,88,97,152,153,154,155,156]. This property allows for measurements of adsorption in situ when the analysis of the spectra makes it possible to distinguish the adsorbed form of the dye from the non-adsorbed form. Although the changes in the spectra after adsorption are often rather small, it is possible to distinguish these forms using special mathematical methods, especially when a larger amount of data is available. One example is methods such as multiple linear regression, which can be used when the spectra of adsorbed and non-adsorbed forms are known. Another example is various chemometric methods that can be used to distinguish similar spectral forms of the dye, such as principal component analysis and multivariate curve resolution–alternating least squares [72,73,74,157,158]. The application of these methods is not so trivial, but they offer a promising direction for the development of more accurate chemical analysis. Measurements of in situ adsorption can be performed with rapid measurements using spectral methods. For example, it is possible to use UV-visible spectra measurements in diode array detector mode for adsorption kinetics. In this way, it is possible to make a series of hundreds of measurements of complete spectra over the required wavelength range and reaction time for each reaction. Large data series greatly increase the accuracy of chemometric analysis [73]. Furthermore, with in situ measurements, there are no adverse effects on sample processing when the supernatant is separated from the dye/clay mineral complex. In addition to in situ measurements over time to characterize adsorption kinetics, it is also possible to perform spectrophotometric titrations for the in situ determination of the adsorption isotherm [88]. In such a case, by adding the dye that exceeds the capacity of the adsorbent by exceeding the equivalence point, a spectral form corresponding to the excess dye in the supernatant can be recorded. The spectral characteristics of the free, non-adsorbed form of the dye are generally very similar or identical to those observed for solutions. It is also possible to include luminescence measurements, which are very sensitive to the environment of the molecules and luminophore groups, extending the range of spectroscopy methods that are available. However, photophysical interaction between different spectral species can interfere with the results of quantitative measurements in this case. The main issues summarized in this chapter are shown in the schematic representation in Figure 7.

## 8. Importance of Adsorption Processes from the View of Environmental Issues

So far, thousands of papers have been published dealing with clays and clay minerals as efficient adsorbents for organic dyes to remove these substances as pollutants from wastewater or industrial wastes [7,159]. From an economic point of view, clay raw materials that are readily available from local deposits are the cheapest solution. They can be used in large quantities and their transportation often represents the largest financial cost of using these materials [160,161]. Among clays, bentonites are perhaps the most important raw material for the highly efficient adsorption of organic substances [15], especially organic cations, including various types of organic dyes. The exceptional properties of bentonites result from the adsorption efficiency of smectites (mainly montmorillonite), which are the main constituents of these materials. The permanent charge in the smectite layers predestines them for the highly efficient adsorption of organic cations. The high adsorption capacity of bentonite for organic cations, including cationic dyes [9], is essentially related to the smectite content. The use of bentonites for the adsorption of pollutants is often limited by some unsolved problems, such as the treatment and regeneration of the adsorbent containing toxic organic substances. On the other hand, the concentration of toxic pollutants in the solid of the adsorbents being used facilitates the management of toxic substances and opens up new avenues for waste processing [2].

Nowadays, synthetic or modified materials that are based on clays or clay minerals are increasingly used [162]. They often contain other functional components or groups that are actively involved in adsorption. One of the oldest examples is acid activation. The resulting materials exhibit a greater ability to adsorb neutral macromolecular organic substances [163]. More modern methods include chemical modifications by introducing additional components and/or functional groups. The advantages of such materials are some properties that are not observed in inexpensive raw materials, such as the regeneration and reuse of the adsorbent, the controlled degradation of adsorbed pollutants, etc. The disadvantage is the higher price and also the fact that these materials contain other components that must be produced industrially, and in this way, contribute indirectly to the pollution of the environment. On the other hand, there are also complex materials that contain other available natural substances as effective modifiers or components besides the clay component [164,165]. These include natural biopolymers such as cellulose and chitosan, which themselves have interesting adsorption properties and are biodegradable [166]. More commonly, non-expandable clay minerals or synthetic materials are used and new adsorbents are being developed [3,8,167]. The further development of improved materials for efficient adsorption as well as the degradation of pollutants is expected mainly in the field of fundamental research, but in industrial practice, the use of traditional and available raw materials will continue in the future. The examples of modified clays and more complex systems are analyzed in the next chapter.

## 9. Brief Examples of Adsorption in Hybrid and Complex Systems

Since there are thousands of works in this field, only selected examples from recent years are given in the following section to show the variability of the possible types and examples of clay-related materials.

### 9.1. Organoclays

Organoclays, prepared by modifying clay minerals with organic cations, have already shown exceptional properties in adsorbing various organic substances. Quaternary ammonium or phosphonium salts are usually used for the modification. Especially in aqueous environments, an agglomerate of organoclay particles is formed as a hydrophobic phase that can be easily separated from the polar aqueous phase. The adsorption of organic molecules from the aqueous phase is thermodynamically advantageous due to the hydrophobic effect. The intercalation of organic molecules is thermodynamically favored due to the weak interactions between the alkyl chains of the surfactants and the siloxane surface of organoclays. Therefore, organoclays are efficient adsorbents for cationic, neutral, and anionic dyes [168]. An example is the adsorption of the anionic dye Allura Red on bentonites modified with the cationic surfactant hexadecyl trimethylammonium and dimethyl dialkylammonium cations [169]. When the degree of modification with organic cations is high, there is also a charge change from negative to positive, which further supports the adsorption of anions. However, the materials still retain the ability to adsorb cationic dyes. The adsorption of anionic dyes in the presence of cationic dyes can be interpreted similarly. Although anionic dyes are not readily adsorbed on the surface of clay minerals with a negative charge, a relatively efficient adsorption of anionic dyes from the mixture of cationic and anionic dyes was observed [168,170,171]. This was explained by the hydrophobic interaction within the organic phase. However, the mechanism that is based on the formation of ion pairs and their subsequent adsorption cannot be neglected either.

### 9.2. Grafted Clays

Although the basal surface of the particles constitutes a significant part of the active surface, the edges of the particles are chemically much more variable, their charge and properties depend on pH and, above all, they open up new possibilities for further chemical modifications. Through a targeted reaction, it is possible to modify the edges of the particles to create particles that have edges with locally hydrophobic properties or with groups with a positive or negative charge [136,171,172,173]. In addition, the edges can be variably modified while maintaining the original properties of the siloxane surface. Edge modifications are often used to achieve specific surface properties. An example is the use of organochlorosilanes to graft bentonite to obtain materials for the adsorption of Sudan dyes [174]. Silylation can be performed in combination with microwave activation [175]. The procedure was applied to the modification of bentonite with cationic silane and the resulting material was efficient for the adsorption of the anionic dye Reactive Violet 5R [175].

### 9.3. Modification with Polymers

Complexes of clay minerals and polymers as well as clay/polymer nanocomposites represent a very wide range of materials with different chemical and physical properties and structures. Complexes with polymeric substances modify the surface properties of clay particles, similar to modification with smaller organic molecules. Polycations, like cationic surfactants, cause a charge reversal that is good for the adsorption of anionic or some neutral dyes [176]. Another example is the complex material of bacterial cellulose and Ca^2+^-montmorillonite with a microporous structure, which was effective in the adsorption of MB and tetracycline from wastewater [177]. Similar materials were recently developed for crystal violet adsorption [178]. Clay/polymer composites have advantages over simple clay minerals in terms of the processability and separation of polymer composite particles from wastewater and dye solutions [179]. There are a variety of applications for these materials. It is worth mentioning an example of efficient sorbents based on nanocomposites of poly(methyl methacrylate) and organoclay [180]. Organoclays, which are part of nanocomposites, can remain active in the adsorption of organic molecules and retain their properties even in composites with polymers, as was shown for the composites with alginate [169]. Microfiber membranes based on a composite of montmorillonite, chitosan, and poly(vinyl alcohol) prepared via electrospinning were used for wastewater treatment [181]. Only a low % of montmorillonite was sufficient to obtain membranes with a high affinity and ultrafast adsorption of organic dye molecules. It was also found that the prepared membranes can be reused for multiple adsorption/regeneration cycles [181]. Complexes with adsorbed dyes can be used as active antibacterial materials [182]. Some types of hydrogels are a particular type of clay/polymer composites. Pseudorotaxanes were formed from amino-modified cyclodextrins attached to poly(propylene glycol) bis(2-aminopropyl ether) chains and were used to modify synthetic Laponite [183]. They were efficient in adsorbing the cationic dyes rhodamine B, crystal violet, and MB. Hydrogels of bentonite and polyvinyl alcohol exhibited similar properties [184]. Other hydrogels were efficient in adsorbing cationic dyes [185] as well as the anionic dye Congo red, which is not readily adsorbed on non-modified bentonite [186]. Similar materials have been further improved by forming pores in cellulose/montmorillonite composites [187,188] including aerogels [189]. Gelatin is another biopolymer that forms stable hydrogels with clay minerals and efficiently adsorbs several types of organic dyes [190].

### 9.4. Magnetic Materials

Magnetic particles are promising because they provide a simple method for separating an adsorbent from wastewater. Iron salts have been used to modify clay minerals for this purpose. One example is a hybrid material with magnetic properties used in the removal of malachite green [191]. A similar approach was described for the adsorption of Congo red on a magnetite/bentonite hybrid material [192], or reactive Yellow 81 on a magnetically separable Fe_3_O_4_/organo-montmorillonite [193]. Complex magnetic oxide nano-porous clay materials were developed for the removal of acid fuchsin [194]. Hydrogels with bentonite and magnetic particles were also efficient for pesticides and organic dyes [195]. Because of its importance for future applications, this topic has been recently reviewed [196,197].

### 9.5. Multicomponent Materials and Complex Systems

In some cases, complex materials exhibit superior properties with added value compared to their constituents. For example, hybrid materials of montmorillonite and graphene or graphene oxide showed a better adsorption of MB [75]. The addition of only a low % of the chitosan polyelectrolyte to the organoclays resulted in a multiplication of the absorption capacity and adsorption rates [198]. It was explained by the mediating effect of anionic dyes that linked cationic components—polyelectrolyte molecules and surfactant cations—on the surface of organoclay particles, allowing for the formation of complex hybrid structures. Another example is materials based on bentonites that were functionalized via silylation under microwave heating and subsequently reacted with glutaraldehyde [199]. The resulting materials showed a good adsorption of violet 5R, giving rise to new pigments with potential industrial applications. The aerogel with a three-dimensional structure, based on a composite of reduced graphene oxide and montmorillonite, was able to treat several pollutants in wastewater, including the dyes [200]. It proved to be very efficient in removing MB and Cr^VI^. Anionic dyes are well adsorbed on some layered materials whose particles carry a positive charge. The general-purpose materials adsorbing both anionic and cationic dyes can be prepared by combining layered particles with both charges. An example is Ni and Fe-layered double hydroxide combined with montmorillonite [201]. Hybrid nanoflocks prepared using a one-pot hydrothermal method showed efficient adsorption of both a cationic and an anionic dye. The growth of smectite layers on the surface of spherical silica micro-particles with subsequent grafting with octyl chains led to the formation of complex systems exhibiting the adsorption of anionic as well as cationic dyes, phloxine B and MB [202]. In recent years, the complexes of metal–organic frameworks with clay and clay minerals have been developed, exhibiting a superior ability to absorb organic dyes [203,204,205,206]. Such materials benefiting from the advantageous properties of both component types have not been studied extensively yet.

Very complex and advanced materials are not always the best choice for the adsorption of dyes from wastewater for economic reasons. Very often, it is possible to find much simpler, more accessible, and, above all, cheaper sorbents, such as bentonite, a very effective sorbent for all soluble cationic dyes. On the other hand, the adsorption of dyes in complex systems can be of great importance for the development of new multifunctional materials, in which dyes play an important and specific role due to their photophysical, photochemical, and optical properties. An example is the composites of bentonite and chitosan that are used as adsorbents for natural aqueous extracts of sappan wood [207]. The adsorbed dye was stabilized in the complex, in comparison to an unstable pure extract. The increase in stability is an important factor for the use of such complex materials in industry [207].

Multicomponent systems that have multiple functionalities can offer several advantages. In addition to effective adsorption, other components can provide the catalytic degradation of adsorbed contaminant molecules. Since hundreds of articles have already been published on this subject, only a few examples will be mentioned: hyperbranched polyester nanocomposites with different doses of bentonite and carbon dots were synthesized via in situ polymerization [208]. In addition to adsorption, the nanocomposite was also used as a photocatalyst that sensitized the photodestruction of organic dyes. On the other hand, the nanocomposite was biodegradable, making it one of the most promising candidates for modern ecological, yet cheap and efficient, adsorbents [208]. Another example is chitosan–montmorillonite composites combined with TiO_2_. The complex removes methyl orange via synergistic adsorption and photodecomposition processes [209]. Silver is another example of a component for clay-based photocatalysts [210].

## 10. Conclusions

The adsorption of organic dyes on clay minerals is one of the most common topics addressed in various disciplines, such as environmental science and the development of new sorbents and materials. Adsorption analysis is often performed according to commonly recurring schemes that do not always reflect the complex nature of this subject. The complex structure and variable properties of clay minerals and the possible influence of impurities in clays are often underestimated. These factors are rarely considered when the model for the adsorption mechanism is proposed. Moreover, adsorption can often be accompanied by major changes in the dye itself, such as a change in the structure, reactions, and molecular aggregation, which is naturally reflected in the adsorption mechanism. In addition to the numerous theoretical models that are defined by various functional dependencies, it is also necessary to consider the diffusion processes that are an essential part of such systems. These include interparticle and intraparticle diffusion, the mechanisms of which are quite complex even in simple systems based on organic dyes and stable clay mineral colloids. The complexity increases when adsorption is accompanied by changes in the colloidal system, such as the agglomeration of clay particles. In addition to the complexity of defining a theoretical model, there are also problems in obtaining correct and accurate adsorption values. Not all possibilities of using the modern methods, procedures, and high-quality processing of the obtained data are used. Despite all the problems, in practice, there are currently enough materials based on clays and modified clay minerals that can effectively adsorb various types of organic pollutants, including organic dyes. Materials with selective adsorption of the target pollutant are challenging. Selectivity for a pollutant should apply in environments with a mixture of different chemical substances, whose concentrations can often exceed the concentration of the pollutant. In addition to selective adsorption, the future will include complex systems that, in addition to adsorption, are capable of efficiently breaking down pollutants into harmless products in a controlled manner, ideally leading to the mineralization of the pollutants. The future will belong to materials that, in addition to effective adsorption, also ensure the catalytic degradation of pollutants, while these materials do not represent a burden to the environment. The development of such materials will require novel strategies and a multidisciplinary approach.

## Figures and Tables

**Figure 1 molecules-28-06951-f001:**
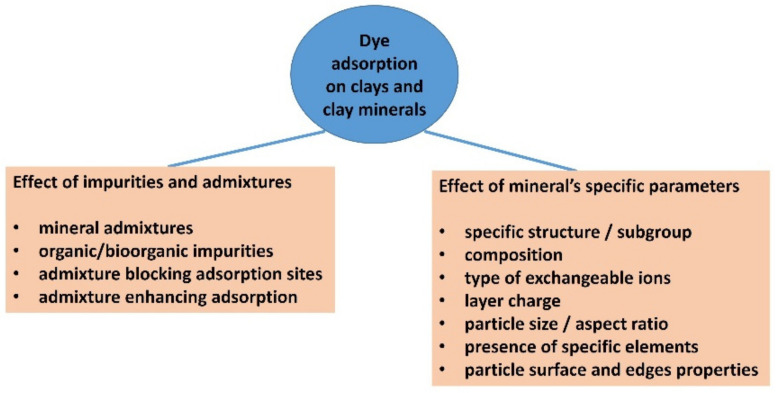
A scheme summarizing the main possible problems and significant factors of dealing with the adsorption of organic dyes on clay minerals, related to insufficient characterization or underestimating the complexity of a clay component.

**Figure 2 molecules-28-06951-f002:**
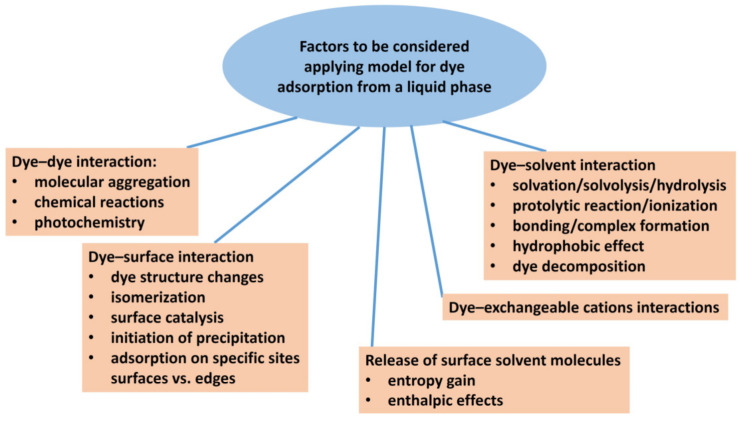
Scheme showing various factors that may influence the adsorption of organic dyes on clay minerals, which can have a decisive influence on the choice of a suitable physical model describing the adsorption isotherm.

**Figure 3 molecules-28-06951-f003:**
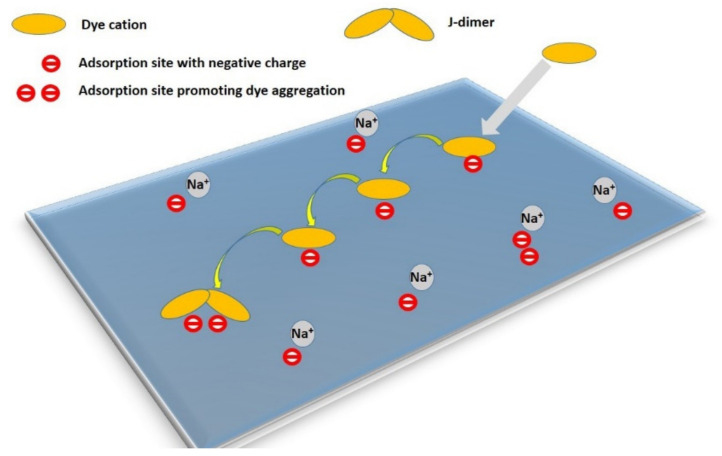
The mechanism of the formation of molecular aggregates proceeding via initial adsorption, desorption–re-adsorption combined with the diffusion of the cations, and reaching a suitable site for the formation of a molecular aggregate. Reproduction of material from PCCP (Physical Chemistry Chemical Physics) reproduced from ref. [73]: Šimonová Baranyaiová, T.; Mészáros, R.; Sebechlebská, T.; Bujdák, J. Non-Arrhenius kinetics and slowed-diffusion mechanism of molecular aggregation of a rhodamine dye on colloidal particles. *Phys. Chem. Chem. Phys.*
**2021**, *23*, 17177–17185, doi:10.1039/d1cp02762j, with permission from the PCCP Owner Societies.

**Figure 4 molecules-28-06951-f004:**
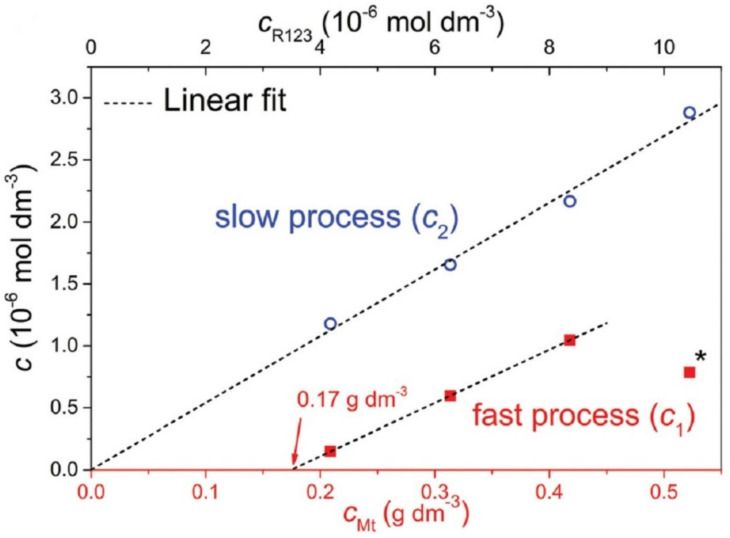
The dependence of the concentration of rhodamine 123 (R123) cations that were involved in the fast and slow process of the molecular aggregation of the dye on the concentration of montmorillonite (Mt) particles. The initial dye/Mt ratio was always constant. The slow process occurs even at very low Mt concentrations when the distance between the Mt particles is very large. This is an indication that the slow process is related to the lateral or intraparticle diffusion of R123 molecules. On the other hand, the fast process does not occur at very low Mt concentrations, but its minimum value is required. The fast process is related to interparticle diffusion, which cannot be active at dilutions when the interparticle distance is too large. Reproduction of material from PCCP (Physical Chemistry Chemical Physics), reproduced from ref. [73]: Šimonová Baranyaiová, T.; Mészáros, R.; Sebechlebská, T.; Bujdák, J. Non-Arrhenius kinetics and slowed-diffusion mechanism of molecular aggregation of a rhodamine dye on colloidal particles. *Phys. Chem. Chem. Phys.*
**2021**, *23*, 17177–17185, doi:10.1039/d1cp02762j, with permission from the PCCP Owner Societies.

**Figure 5 molecules-28-06951-f005:**
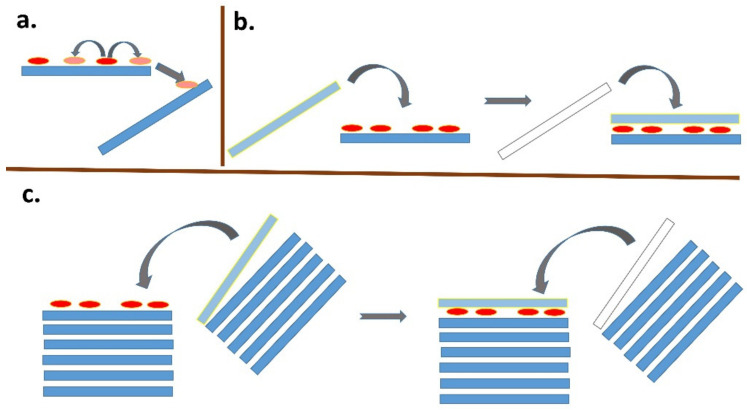
Schematic representation of various phenomena occurring after initial adsorption. (**a**) Lateral and interparticle diffusion processes. (**b**) Destabilization of the surface of the colloidal particles by the organic dye molecules that are present leads to agglomeration with another particle. A high dye-to-particle ratio is required for this phenomenon. (**c**) Same as (**b**), but starting from particle agglomerates. Adsorbed dye molecules destabilize the outer surface, leading to association with another particle. A relatively low dye-to-clay ratio is sufficient for this phenomenon to occur.

**Figure 6 molecules-28-06951-f006:**
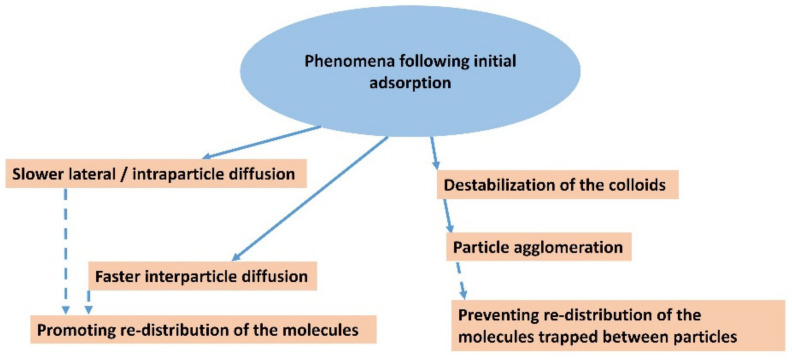
Schematic representation showing the main possible phenomena that may occur after the initial adsorption of dye molecules.

**Figure 7 molecules-28-06951-f007:**
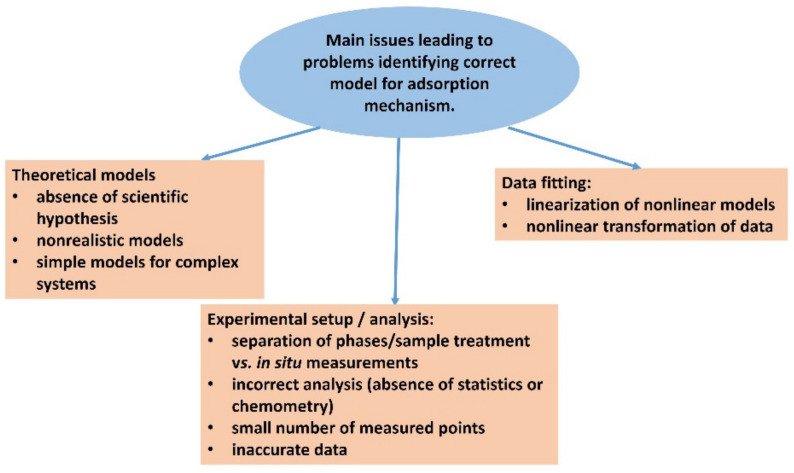
Scheme summarizing the main problems in the identification of the correct theoretical model for adsorption. It relates to the proper selection of the model, correct experimental setup, analysis, and data processing.

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
