# Peer review of "Controversial Issues Related to Dye Adsorption on Clay Minerals: A Critical Review"

_molecules, 2023, doi:10.3390/molecules28196951_

Round 1

Reviewer 1 Report

The manuscript (Ref. No. 2614950) is offered as a critical review of controversial issues related to dye adsorption on clay minerals. The author has tried to summarize and analyze information about problematic issues found in publications on the adsorption of dyes on clay minerals. Overall, the subject is interesting to readers who specialize in adsorption pollutant removal from water. I believe that the manuscript can be published in Molecules, but a minor revision is recommended before publication.

Some of my comments are as follows:

1)       It should be removed "eq" in the numbering of equations.

2)       It should be checked in the equation 9.

3)       It should be an explanation of all the quantities included in the equations.

4)       There is only information about adsorption models and adsorption kinetics. It should provide controversial information about thermodynamic studies.

Author Response

Reviewer comments:

  1. It should be removed "eq" in the numbering of equations.

2)         It should be checked in the equation 9.

3)         It should be an explanation of all the quantities included in the equations.

The response to the comment:

The equation numbering format was corrected to follow the suggestions of the reviewer. The explanations of all the symbols were added and checked again for completeness. 

Reviewer comments:

4)         There is only information about adsorption models and adsorption kinetics. It should provide controversial information about thermodynamic studies.

The response to the comment:

Thermodynamic studies are often based on simplified models for very complex materials with a broad variety of adsorption sites and interactions. The complex character of the systems based on clays and related materials makes the simple models inappropriate for modeling the adsorption. As a result, the thermodynamic studies using these simplified models cannot describe the real state of the systems, so provide false results. Some aspects of this issue are indicated and discussed in the 2nd section of the review “Theoretical basics” and Figure 2.

Reviewer 2 Report

This review presents the issues and problems that current studies of the adsorption of organic dyes on clay minerals have had. The author discussed the complexity of the adsorption such as effects of impurities, diffusion, changes of adsorbates and adsorbents, etc. that theoretical models current studies have used could not account for. The discussion is detailed, thorough, and hence insightful. The manuscript was well-written, and I suggest publishing it in Molecules. I only have a couple of minor comments.

1) The diffusion of organic molecules into the interlayers between alumino-silicate layers is missing or needs a little more attention. This diffusion also contributes to the complexity of the adsorption. It would be great if the author could add more to that.

2) It would also be valuable if the author envisages a little more in the future.

3) Change ‘if’ to ‘of’ in the low-left panel of Figure 1.

4) Remove label b at the upper-left corner of Figure 4.

Author Response

Reviewer comments:

1) The diffusion of organic molecules into the interlayers between alumino-silicate layers is missing or needs a little more attention. This diffusion also contributes to the complexity of the adsorption. It would be great if the author could add more to that.

2) It would also be valuable if the author envisages a little more in the future.

The response to the comment:

I added more than one page on the intercalation of dye molecules between the layers (The end of section 6.). This subtopic is so broad it would require a deep individual analysis and another review article. Therefore, only basic concepts are mentioned in this paragraph together with references to some representative pieces of the literature. It is impossible to go deeper into this topic since it would expand the present manuscript too much losing the linkage to the main message of this work. 

Reviewer comments:

3) Change ‘if’ to ‘of’ in the low-left panel of Figure 1.

4) Remove label b at the upper-left corner of Figure 4.

The response to the comment:

The figures were corrected as suggested by the reviewer.

Reviewer 3 Report

The paper is concerned with the Controversial issues related to dye adsorption on clay minerals: A critical review. The paper covers some important issues but some points should be corrected. I recommend the manuscript for publication after major revision.

2. Page 1: Abstract: Need to revise properly. Add results, conclusion, and prospects of this review. Confirm the word limit for the abstract.

3. Only a few articles from recent years, therefore, need to add the most recent reference in the application section.

4. In the introduction, it should be discussed detailed about the difference of this material with other materials.

5. Please improve the conclusive remarks after each section. Although they are present, but can be improved a bit.

7. The resolution of images must be improved during the revision stage.

8. What potential does further research hold? What is the ultimate goal in this field?

9. The section of Conclusion and outlooks is too simple; the authors should give deeper insights into the advantages, loopholes and future development direction of MOF networks.

10. Does the future of study lie in this area? Are there other more promising areas in the field which could be progressed?

11. Fig 1: There is no discussion in the text in detail

12. Need to add the exact sensing mechanisms for each sensing application

13. Abbreviations must be defined at their first mention (abstract and text separately).

check

Author Response

Reviewer’s comments:

  1. Page 1: Abstract: Need to revise properly. Add results, conclusion, and prospects of this review. Confirm the word limit for the abstract.

The response to the comment:

To be within the limit of 200 words, some sentences were simplified and the prospects were briefly explained at the end of the abstract.

Reviewer’s comments:

  1. Only a few articles from recent years, therefore, need to add the most recent reference in the application section.

The response to the comment:

A deeper analysis of the literature published in recent year was done and new references and text were added.

Reviewer’s comments:

  1. In the introduction, it should be discussed detailed about the difference of this material with other materials.

The response to the comment:

The focus of the review article is already broad counting over 18000 words, targeting various aspects of research on the adsorption of organic dyes to clays and clay minerals. It is true that many of the issues also arise in work dealing with the adsorption of other substances or other sorbents, and thus some parts of the article might be of use to a broader readership. On the other hand, expanding this already extensive review article would be counterproductive. Therefore, I have decided not to expand the text with another section comparing different types of materials.

Reviewer’s comments:

  1. Please improve the conclusive remarks after each section. Although they are present, but can be improved a bit.

The response to the comment:

It has been done. I believe, in particular, schematic illustrations in some figures can provide the readers with easily understandable summaries of the chapters.

Reviewer’s comments:

  1. The resolution of images must be improved during the revision stage.

The response to the comment:

The resolution was improved to follow the guidelines of the journal.

Reviewer’s comments:

  1. What potential does further research hold? What is the ultimate goal in this field?

The response to the comment:

Clay minerals are natural materials with a favorable price and at the same time with an infinite potential for the development of modified and complex materials with new functionalities. The problems of environmental protection will put more and more demands on new technologies for dealing with pollutants, and new adsorbents will be a part of these technologies. Clay minerals can be one of the affordable precursors of modern materials that, after appropriate modification, can achieve high benefits at relatively low cost.

Reviewer’s comments:

  1. The section of Conclusion and outlooks is too simple; the authors should give deeper insights into the advantages, loopholes and future development direction of MOF networks.

The response to the comment:

The conclusion was partially modified. MOF complexes with clays are also mentioned in the section dealing with complex materials based on clays.

Reviewer’s comments:

  1. Does the future of study lie in this area? Are there other more promising areas in the field which could be progressed?

The response to the comment:

These are very general questions that are very difficult to answer. The development of scientific disciplines often goes in directions that no one foresaw. This is also the case with clay minerals. In the past, they represented traditional materials, but today they are becoming part of hybrid materials with surprising applicability in modern industry.

Reviewer’s comments:

  1. Fig 1: There is no discussion in the text in detail

The response to the comment:

The discussion was expanded, in particular the issues related to the role of parameters of clay minerals.

Reviewer’s comments:

  1. Need to add the exact sensing mechanisms for each sensing application

The response to the comment:

This is out of the scope of this review article.

Reviewer’s comments:

  1. Abbreviations must be defined at their first mention (abstract and text separately).

The response to the comment:

The text was checked carefully to follow the rules.

Round 2

Reviewer 3 Report

accept